# Providing Scale to a Known Taxonomic Unknown—At Least a 70-Fold Increase in Species Diversity in a Cosmopolitan Nominal Taxon of Lichen-Forming Fungi

**DOI:** 10.3390/jof8050490

**Published:** 2022-05-08

**Authors:** Yanyun Zhang, Jeffrey Clancy, Jacob Jensen, Richard Troy McMullin, Lisong Wang, Steven D. Leavitt

**Affiliations:** 1Key Laboratory for Plant Diversity and Biogeography of East Asia, Kunming Institute of Botany, Chinese Academy of Sciences, Heilongtan, Kunming 650201, China; 2021209@ahnu.edu.cn; 2College of Life Science, Anhui Normal University, Wuhu 241000, China; 3Department of Biology, Brigham Young University, 4102 Life Science Building, Provo, UT 84602, USA; jclancy1330@gmail.com (J.C.); jnjensen88@gmail.com (J.J.); 4Canadian Museum of Nature, Research and Collections, Ottawa, ON K1P 6P4, Canada; tmcmullin@nature.ca; 5Department of Biology, M. L. Bean Life Science Museum, Brigham Young University, 4102 Life Science Building, Provo, UT 84602, USA

**Keywords:** alpine/arctic/Antarctic, ASAP, cosmopolitan, cryptic species, genome skimming, species delimitation, symbiotic phenotype

## Abstract

Robust species delimitations provide a foundation for investigating speciation, phylogeography, and conservation. Here we attempted to elucidate species boundaries in the cosmopolitan lichen-forming fungal taxon *Lecanora polytropa*. This nominal taxon is morphologically variable, with distinct populations occurring on all seven continents. To delimit candidate species, we compiled ITS sequence data from populations worldwide. For a subset of the samples, we also generated alignments for 1209 single-copy nuclear genes and an alignment spanning most of the mitochondrial genome to assess concordance among the ITS, nuclear, and mitochondrial inferences. Species partitions were empirically delimited from the ITS alignment using ASAP and bPTP. We also inferred a phylogeny for the *L. polytropa* clade using a four-marker dataset. ASAP species delimitations revealed up to 103 species in the *L. polytropa* clade, with 75 corresponding to the nominal taxon *L. polytropa*. Inferences from phylogenomic alignments generally supported that these represent evolutionarily independent lineages or species. Less than 10% of the candidate species were comprised of specimens from multiple continents. High levels of candidate species were recovered at local scales but generally with limited overlap across regions. *Lecanora polytropa* likely ranks as one of the largest species complexes of lichen-forming fungi known to date.

## 1. Introduction

Quantifying the full scope of species diversity is perhaps one of the largest undertakings of modern taxonomy [1]. Form and function have historically played fundamental roles in inferring species boundaries and relationships among species [2]. More recently, the incorporation of genetic information has highlighted limitations in past attempts to characterize the earth’s biological diversity based exclusively on phenotypes [3]. While the science of naming, describing, and classifying organisms has fallen largely to taxonomists, the implications of taxonomy extend to questions of the origin of novelty, speciation, symbiosis, conservation, and ecology [4,5,6]. Complementary, modern taxonomists use an ever-expanding toolbox from other disciplines for systematic and taxonomic research, integrating traditional approaches with genetic and ecological data, computational modeling, and empirical species delimitation [7,8,9,10].

As with other organismal groups, this integrative taxonomic approach has transformed our understanding of fungal diversity—challenging current taxonomy at multiple levels and highlighting rampant unrecognized diversity [11,12]. Though underestimates and challenges in establishing accurate inventories of fungal diversity may be expected due to the mostly cryptic lifestyle of fungi, reconsidering diversity in well-known, conspicuous fungi has also led to the realization that species diversity in these groups may also be mischaracterized [13,14,15,16]. Traditionally, and often due to lack of available genetic data, fungal species boundaries have been delimited based largely on morphological characteristics as they can be incredibly practical, especially in the field. However, robust phenotype-based species delimitations in fungi are often confounded by the existence of morphologically similar species in which intra- and inter-specific variations overlap regarding some characters that have been traditionally used to separate taxa [17].

Factors relating to fungi in obligate symbioses add another layer of complexity in delimiting species-level lineages. In lichen-forming fungi, stable interactions among multiple symbionts—the fungal nutritional specialists that acquire fixed carbon from photoautotrophs and are supported by other microbes [18,19,20,21]—result in a persistent thallus which can be considered a symbiotic phenotype that results only from the interaction of unrelated organisms [22]. While the symbiotic phenotype of lichens can provide important insight into species boundaries of lichen-forming fungi—the mycobiont, interactions of the mycobiont with differing suites of microorganisms may result in diverse symbiotic phenotypes of the same fungal species [23,24]. In other cases, carefully considering the lichen-associated symbionts can provide crucial insight for resolving otherwise apparently cryptic fungal species [25]. While the complexities of lichen symbioses and their implications for interpreting lichen-forming fungal diversity have recently been more fully recognized, commensurate advances in practical approaches to empirically delimiting lichen-forming fungi have also occurred [26].

Ongoing advances in sequencing technologies have been key in revealing the presence of previously unknown species diversity in fungi [12], opening the gates for research into lichen symbioses that would have previously been impossible [27,28,29,30]. Research focusing on long-standing, well-known problematic groups has been shown to lead to crucial resolution. For example, detailed investigations of a widespread montane, tropical macrolichen representing a single nominal fungal species highlighted that the species diversity of the name-giving mycobiont had been underestimated by orders of magnitude [16,31]. The question remains, is this vast underestimate of species-level diversity in lichen-forming fungi an anomaly or a more widespread phenomenon across lichen fungi?

*Lecanora polytropa* (Hoffm.) Rabenh. (Figure 1)—the name-giving mycobiont for the associated symbiotic phenotype—is distributed across largely disjunct, intercontinental populations and occurs on siliceous rocks (especially granite) in montane, alpine, and arctic/Antarctic habitats (Figure 2). As the name suggests, *L. polytropa*, and related species, exhibit a wide range of morphological diversity (Figure 1). Morphological diversity across the name-giving nominal taxon has been recognized over the course of two centuries of research, with dozens of described forms or varieties. At the same time, limited studies have explored the relationship of *L. polytropa* with other *Lecanora* species [32,33,34,35,36], and the full extent of diversity within the ‘*L. polytropa* group’ itself remains largely uncharted. Delimiting species boundaries is made more difficult by the potential role that the environment plays in shaping morphology. Specimens of *L. polytropa* that grow near copper mines, for instance, often have their color change from the normal yellow-green to turquoise blue [37]. While careful phenotypic assessments have led to the description of some species within the ‘*L. polytropa* group’ [32,33], the notorious variability of this species group, coupled with the occurrence of intermediate morphotypes within the ‘*L. polytropa* group’ [38] highlights the pressing need for a thorough survey of this group.

To better understand the scale of unrecognized species-level diversity in this common, cosmopolitan lichen-forming fungal species, here we circumscribe candidate species within this complex using DNA sequence data. We sampled over 300 specimens collected across multiple, intercontinental populations to meet this aim. From these specimens, we generated DNA sequence data from the standard DNA barcode marker—the internal transcribed spacer region [39]. A subset of specimens was then selected to represent the genetic diversity observed from the ITS data to generate multi-locus and genome-scale datasets. Based on these data, we provide compelling evidence that species diversity within the ‘*L. polytropa* group’ is vastly underestimated, with the nominal taxon *L. polytropa* representing at least 70 candidate species based on current, limited sampling and additional unrecognized species diversity in other species within the ‘*L. polytropa* group’. In so doing, this study lays a transformative framework for future studies to characterize species diversity more fully within the ‘*L. polytropa*’ species complex.

## 2. Materials and Methods

### 2.1. Taxon Sampling

Our sampling targeted specimens in the ‘*L. polytropa* group’ [35,40]. Recently, the ‘*L. polytropa* group’ was found to be a major lineage within the provisionally named ‘MPRPS clade’ *sensu* [35] within Lecanoraceae (comprising *Myriolecis*, *Protoparmeliopsis*, *Rhizoplaca*, the ‘*L. polytropa* group’, *Bryonora*, and the “*Lecanora*” *saligna* group). For this study, sampling efforts focused on *L. polytropa sensu lato* populations in Asia, Europe, and North America, supplemented with ITS sequences from GenBank, including *Lecanora chlorophaeodes* Nyl. (*n* = 3), *L. dispersoareolata* (Schaer.) Lamy (1), *L. fuscobrunnea* Dodge & Baker (31), *L. intricata* (Ach.) Ach. (4), *L. polytropa* (22), *L. solaris* Yakovchenko & Davydov (7), *L.* cf. *subcinctula* (Nyl.) Th. Fr. (2), *L. subintricata* (Nyl.) Th. Fr. (19), *Rhizoplaca aspidophora* (Vain.) Redón (1), and unidentified sequences also recovered within the ‘polytropa group’ clade (5) (Figure 2). The genus *Carbonea* was shown to be closely related to the ‘polytropa group’ [40], and all 21 ITS sequences currently available on GenBank were included here. Ultimately, a total of 340 specimens were included (Appendix A). We note that in several cases, multiple lichen thalli were selected from the same vouchered collection when the voucher included multiple, distinct *L. polytropa sensu lato* thalli. To assess the range of diversity at a limited geographic scale in this nominal taxon, ‘*L. polytropa* group’ specimens were relatively densely sampled from the La Sal Mountains—a sky island on the Colorado Plateau, Utah, USA. Other geographic regions were not sampled as densely, aiming to characterize broader-scale patterns of distributions of putative species-level lineages. We could not obtain fresh material from alpine/subalpine habitats in Africa, Antarctica, Australia, Central America, or New Zealand, and only limited sequences from South America were available [35].

To explore the potential for diagnostic phenotypic traits separating specimens for distinct candidate species-level lineages circumscribed in this study, we characterized (i) general growth forms, (ii) spore sizes, and (iii) secondary metabolites. Morphological characters—e.g., thallus characters, surface color/texture, apothecial disk color, apothecia margin, etc.—were assessed using an Olympus SZH dissecting microscope. Observations and measurements of ascospores were made in water with an Olympus BH-2 microscope, with multiple ascospores measured from at least two apothecia on each specimen. Chemical constituents were identified using thin-layer chromatography (TLC), following standard methods with solvent systems ‘C’ and ‘G’ [41,42].

### 2.2. DNA Extraction and Sequencing

Total genomic DNA was extracted from specimens collected for this study using the ZR fungal/bacterial DNA miniprep kit (Zymo Research, Irvine, CA, USA), the Wizard Genomic DNA Purification Kit (Promega, Madison, WI, USA), or the DNAsecure Plant Kit (Tiangen Biotech, Beijing, China). For all specimens, we attempted to generate sequence data from the internal transcribed spacer region (ITS)—the standard DNA barcoding marker for fungi [39]. For selected specimens representing the genetic diversity observed from the ITS dataset, for example, attempting to represent as many candidate species-level lineages as possible, including multiple representatives for candidate species, if available (see Results), we targeted four additional loci traditionally used in phylogenetic analyses of Lecanoraceae [40]—a portion of the nuclear large-subunit (nuLSU), a fragment of the gene encoding the mitochondrial small subunit (mtSSU), and fragments from two nuclear protein-coding loci, the RNA polymerase II subunit 1 (*RPB1*) and RNA polymerase II subunit 2 (*RPB2*). Temperature profiles for polymerase chain reaction (PCR) amplification for all loci follow previous studies [40]. PCR amplifications were performed using Ready-To-Go PCR Beads (GE Healthcare, Pittsburgh, PA, USA); or alternatively, in 25 μL reactions containing 12.5 μL 2 × Taq PCR Mix (Tiangen Biotech, Beijing, China), 0.5 μL of each primer, 10.5 μL ddH2 O and 1 μL of DNA. PCR products were visualized on 1% agarose gel and cleaned using ExoSAP-IT (USB, Cleveland, OH, USA), following the manufacturer’s recommendations. We sequenced complementary strands with the same primers used for PCR amplification, and sequencing reactions were performed using BigDye 3.1 (Applied Biosystems, Foster City, CA, USA). Products were run on an ABI 3730 automated sequencer (Applied Biosystems) at the DNA Sequencing Center at Brigham Young University, Provo, UT, USA.

Single marker and multi-locus approaches may be insufficient to robustly delimit species boundaries, particularly among closely related species [43,44]. Genome-scale data provide unprecedented insight into species boundaries and testing concordance among independent loci [45,46,47]. Therefore, in addition to Sanger sequencing of traditional phylogenetic markers, metagenomic reads were newly generated from 32 specimens from the ‘*L. polytropa* group’ using short-read shotgun sequencing [48]. Using metagenomic reads from the selected specimens, independent DNA datasets were assembled to investigate concordance among the species partitions inferred from the ITS markers with clades inferred from genome-scale nuclear and mitochondrial datasets [45]. Specimens for Illumina sequencing were selected based on genetic diversity initially observed from sampling in western North America; as the project expanded, we could not include specimens representing additional diversity observed in subsequent broad geographic sampling for Illumina sequencing. For the specimens selected for metagenomic high-throughput sequencing, total genomic DNA was extracted from a small portion of lichen thalli (comprised of the mycobiont, photobiont, and other associated microbes) using the E.Z.N.A. Plant DNA DS Mini Kit (Omega Bio-Tek, Inc., Norcross, GA, USA) and following the manufacturers’ recommendations. Total genomic DNA was prepared following the standard Illumina whole genome sequencing (WGS) library preparation process using Adaptive Focused Acoustics for shearing (Covaris, Sydney, Australia), followed by an AMPure cleanup step. The DNA was then processed with the NEBNext Ultra™ II End Repair/dA-Tailing Module end-repair and the NEBNext Ultra™ II Ligation Module (New England Biolabs, Ipswich, MA, USA) while using standard Illumina index primers. Libraries were pooled and sequenced with the HiSeq 2500 sequencer in high output mode at the DNA Sequencing Center, Brigham Young University, Provo, UT, USA, using 250 cycle paired-end (PE) reads.

### 2.3. Short-Read Processing and Data Assembly

Raw reads were trimmed using Trimmomatic v0.39 [49] to remove adapter and primer sequences and low-quality reads. Bases at the start and end of reads were trimmed when they had a quality below 3 and 10, respectively, and when the quality of 5-bp sliding windows was <20. All trimmed reads <36 bp were filtered out. We performed a de novo genome assembly using PE reads from *L. polytropa* specimen “Leavitt 16-650” using SPAdes [50]. To identify single-copy nuclear genes for phylogenomic reconstructions from the assembled mycobiont contigs, we used Benchmarking Universal Single-Copy Orthologs to extract up to 1438 gene regions (BUSCO; [51]. Assembled contigs were analyzed using the BUSCO pipeline implemented in the Cyverse.org Discovery Environment [52,53]. The Fungi Odb10 dataset was used to identify BUSCO genes from the assembled *L. polytropa* contigs. Exploratory BLAST searches and assessments of relative sequencing coverage were used to infer that the extracted BUSCO genes likely originated from the *L. polytropa* genome and not other co-occurring fungi. Partial and multi-copy BUSCOs were excluded, and the remaining filtered, single-copy BUSCO genes were used as targets for bait sequence capture using HybPiper [54] to extract these genes regions from each ‘*L. polytropa* group’ metagenomic sample (e.g., [48]). MAFFT [55] was used to generate alignments for individual BUSCO genes using the default parameters, and the alignment algorithm for each locus was chosen automatically by MAFFT. For each alignment, any sample which had an average completion (assembly length/target length) of <0.20 was removed. Genes with average coverage <75% across all BUSCO genes were removed. Phylogenetic gene trees were reconstructed using maximum likelihood (ML) as implemented by IQ-TREE [56]. The substitution model used for each tree was selected using ModelFinder [57]. To assemble genome-scale data from the mycobiont mitochondrial genome, we identified mitochondrial contigs from the SPAdes assembly using BLAST comparisons [58]. The three longest mitochondrial contigs—representing a total of 87.9 Kb—were used as targets in RealPhy v1.12 [59]. For the RealPhy assembly, we used the following parameters to generate the mitochondrial genome alignment: -readLength 100; –perBaseCov 5; –gapThreshold 0.2.

### 2.4. Candidate Species Delimitation Using the Standard Fungal DNA Barcode

Initial candidate species partitions for the *L. polotropa* group were inferred using Assemble Species by Automatic Partitioning (ASAP) [60] based on the multiple sequence alignment of the standard fungal DNA barcode—ITS [39]. ASAP circumscribes species partitions using an implementation of a hierarchal clustering algorithm based on pairwise genetic distances from single-locus sequence alignments [60]. The pairwise genetic distances are used to build a list of partitions ranked by a score, computed using the probability of groups to define panmictic species. ASAP provides an objective approach to circumscribe relevant species hypotheses as a first step in the process of species delimitation. ITS sequences generated for this study were combined with those GenBank and aligned using the program MAFFT v7 [55,61]. We implemented the G-INS-i alignment algorithm and ‘1PAM/K = 2’ scoring matrix with an offset value of 0.1, the ‘unalignlevel’ = 0.2, and the remaining parameters were set to default values. The multiple sequence alignment was analyzed using the ASAP Web Server (https://bioinfo.mnhn.fr/abi/public/asap/, accessed on 27 January 2022), with the ‘asap-score’ considered to select the optimal number of species partitions [60]. In addition to ASAP species delimitations, we also implemented a single-locus tree-based species delimitation method—the Bayesian implementation of the Poisson tree process model (bPTP) [62]. A maximum-likelihood (ML) tree was inferred from the ITS alignment using IQ-TREE [56], which was subsequently analyzed using bPTP with 1,000,000 generations and a burn-in of 10%.

### 2.5. Phylogenetic Analyses and Tests of Genomic Concordance

To assess the monophyly of the ‘*L. polytropa* group’ within Lecanoraceae, ITS, and mtSSU sequences from the present study were combined with the relatively comprehensive “2-locus dataset”, including 251 OTUs representing 150 species, originally reported in [40]. Sequences were aligned using MAFFT v7 [55,61], implementing the same parameters described above for ITS. To minimize ambiguities in multiple sequence alignments (MSA), subsequent MSA and phylogenetic reconstructions were restricted to members of the ‘*L. polytropa* group’ and specimens representing the two putative sister lineages inferred from the family-wide, “2-locus dataset”—*Carbonea* species and those representing *L. subintricata* (see Results). For the ‘*L. polytropa* group’ and putative sister groups, the ITS, nuLSU, mtSSU, *RPB1*, and *RPB2* sequences, were aligned using MAFFT v7 as described above. For both the nuLSU and mtSSU alignments, ambiguously aligned regions were excluded using the Gblocks webserver [63], implementing the options for a less stringent selection. A ML tree for the ‘*L. polytropa* group’ was inferred from the concatenated five-marker alignment using IQ-TREE [56]. The concatenated alignment was partitioned by loci, with substitution models selected using ModelFinder [57] and nodal support assessed using 2000 ultra-fast bootstrap replicates [64].

To assess concordance between the candidate species inferred from the standard fungal barcode (ITS) and genome-scale data, we compared ASAP partitions with (i) clades inferred from concatenated single-copy nuclear markers spanning 2.28 Mb, (ii) clades inferred from a mitochondrial alignment spanning 65.5 Kb, and (iii) a phylogenomic approach to species delimitation. Our genome-scale sampling only represented a subset of the candidate species inferred from the ITS data, and comparisons were limited to the 12 ASAP partitions that also had representative samples with short-read data (32 specimens). We reconstructed a phylogeny from a supermatrix comprised of the 1209 single-copy BUSCO markers. Concatenation approaches provide accurate inferences under a range of conditions [65]. We used IQ-TREE v1.6.9 to generate a ML tree from the concatenated BUSCO supermatrix, with nodal support assessed using 1000 ultra-fast bootstrap replicates [64]. We also used SODA [66], an ultra-fast and relatively accurate method for species delimitation, to assess candidate species boundaries inferred from the BUSCO data. SODA uses frequencies of quartet topologies to determine if each branch in a guide tree inferred from gene trees (1209 BUSCO topologies) is likely to have a positive length. It uses the results to infer a new species tree that defines species boundaries. We ran SODA, implemented in ASTRAL [66], with a *p*-value cut-off of 0.001. The mitochondrial topology was reconstructed from the RealPhy alignment using IQ-TREE [56], with the substitution model selected using ModelFinder [57], and nodal support assessed using 1000 ultra-fast bootstrap replicates [64].

## 3. Results

### 3.1. Sequence Data

Newly generated ITS, nuLSU, *RPB1*, *RPB2*, and mtSSU sequences are deposited in GenBank under accession numbers ON179980–ON180462 and ON217582–ON217793. Illumina short reads from the 32 ‘*L. polytropa* group’ specimens are available in the NCBI Short Read Archive (PRJNA823672). Of the 1438 BUSCO genes searched, 1209 complete, single-copy BUSCO genes were recovered (93.5% of all BUSCO groups). The concatenated alignments of the 1209 nuclear BUSCO markers spanned a total of 2.28 Mb. The REALPHY genome skimming approach for generating mitochondrial data resulted in an alignment of 65456 bp. The concatenated five-marker dataset comprised 3891 aligned nucleotide position characters—ITS (*n =* 380; 603 bp MSA), nuLSU (*n =* 122; 843 bp MSA [ambiguous sites removed]), *RPB1* (*n =* 117; 816 bp MSA), *RPB2* (*n =* 105; 828 bp MSA), and mtSSU (*n =* 112; 801 bp MSA [ambiguous sites removed]).

### 3.2. Candidate Species Inferred Using the Standard DNA Barcode (ITS)

The best-scoring ASAP species partitions delimited between 62–103 candidate species within the ‘*L. polytropa* group’—the 103- and 102-species models had the best asap-scores (Appendix A). Specimens identified as *L. polytropa* represented up to 75 distinct species partitions in the ASAP analyses of the ITS alignment. Multiple candidate species were also recovered within the nominal taxa *L. concolor* (up to 2 species partitions), *L. dispersoareolata* (2), *L. intricata* (6), *L. solaris* (2), *L. sommervelli* (2), and *L. subintricata* (5). Six species partitions comprised *Carbonea* sequences—the currently unsettled sister-clade to the ‘*L. polytropa* group’. Based on current sampling, most candidate species partitions are found in geographically limited areas, with only seven species partitions including samples from multiple continents (Appendix A). The tree-based bPTP species delimitation model resulted in 73 candidate species within the ‘*L. polytropa* group’, which were concordant with most of the ASAP species partitions, with ten of the 73 bPTP candidate species combining multiple ASAP partitions from the best-supported models (Appendix A).

### 3.3. Multi-Locus Phylogenetic Inference and Phylogenetic Concordance among Data Sets

In the combined ITS/mtSSU topology, the ‘*L. polytropa* group’ was recovered as monophyletic with strong bootstrap (BS) support within the ‘MPRPS clade’ *sensu* [35]. In the combined ITS/mtSSU topology, *Carbonea* specimens were recovered as monophyletic within the ‘*L. polytropa* group’; and *L. subintricata* specimens were recovered as sisters to the entire ‘*L. polytropa* group’ with weak support (58% BS; Appendix A). In the 5-marker ‘*L. polytropa* group’ topology, including both *Carbonea* and *L. subintricata* specimens, a midpoint rooted topology showed *Carbonea* specimens as sisters to the ‘*L. polytropa* group’, with *L. subintricata* specimens nested within. Given the uncertainty of the sister clade to the ‘*L. polytropa* group’, we opted to show the midpoint root topology (Figure 3), rather than rooted with *L. subintricata* specimens as inferred from the “2-locus dataset”.

In the 5-marker ‘*L. polytropa* group’ topology, the vast majority of candidate species partitions inferred from the ITS MSA using ASAP were recovered as reciprocally monophyletic (Figure 3; Appendix A). However, the candidate species partitions inferred from the ITS MSA were not recovered as reciprocally monophyletic in any of the topologies inferred from four traditional markers individually (nuLSU, *RPB1*, *RPB2*, and mtSSU), although some species partitions were monophyletic in some single-gene topologies. Specimens representing previously described species that occur within the *L. polytropa* clade were generally recovered as monophyletic—*L. concolor*, *L. dispersoareolata*, *L. fuscobrunnea*, *L. solaris*, *L. sommervelli*, and *L. subintricata*; although multiple candidate species were inferred from ITS sequence data in most of these nominal taxa (Figure 3; Appendix A). In contrast, sequences representing *L. chlorophaeodes* and *L. intricata* were not recovered as monophyletic.

The limited nuclear phylogenomic sampling comprising specimens from the western USA unambiguously supported the ASAP candidate species partitions represented by short-read data (Figure 4A). The mitochondrial phylogenomic data also generally supported the ASAP species partitions with one instance of putative mitochondrial introgression (Figure 4B). The SODA species delimitation analyses based on 1209 BUSCO gene topologies consistently further subdivided the candidate species partitions delimited from ITS sequence data using ASAP, delimiting 20 species from the specimens represented by genome-scale data using SODA, in contrast to the 12 ASAP species partitions (Figure 4A).

### 3.4. Assessing Morphological Concordance with Species Partitions

ASAP partitions represented by multiple specimens often comprised polymorphic morphologies. No phenotypic characters were observed that consistently diagnosed any of the ASAP partitions comprised of three or more specimens, although some characters or combinations of characters (e.g., spore size, growth form, secondary metabolite variation, and morphology) loosely corresponded with some distinct species partitions. In a limited number of cases, secondary metabolite concentrations qualitatively varied among specimens in different ASAP partitions. However, given the uneven representation of specimens per candidate species, with many candidate species represented by a very limited number of species, no quantitative comparisons were made. *Lecanora stenotropa* Nyl. is morphologically similar to *L. polytropa* but with a more brownish-green thallus and smaller, narrowly ellipsoid spores. However, based on anatomical observations, no *L. stenotropa* specimens from western North America were sampled, and the relationship of this taxon to other species-level clades in the *L. polytropa* group remains unknown.

## 4. Discussion

Here, molecular sequence data reveal that the well-known, problematic nominal taxon *Lecanora polytropa* likely ranks as one of the largest species complexes of lichen-forming fungi known to date. Based on species delimitations using alignments of ITS sequences, as many as 103 species partitions were circumscribed with the *L. polytropa* clade using ASAP, including ca. 75 candidate species identified as *L. polytropa* and multiple candidate species in other formally described species in the clade. For comparison, perhaps the most species-rich group is the nominal taxon *Dictyonema glabratum*, likely comprising at least 400 distinct species-level lineages [16]. Most other nominal taxa known to mask multiple species-level lineages comprise perhaps up to ten distinct species-level lineages [29,67,68,69,70,71,72]. Overall, these findings corroborated the perspective that some currently circumscribed conspicuous, well-known lichens may harbor spectacular levels of unrecognized species-level diversity [16]. While phenotypic variation within *L. poytropa s. lat.* is well documented and has historically been interpreted to circumscribe dozens of forms or varieties, a widely accepted workable taxonomy has been elusive for this group over the past two centuries. Below we discuss important implications relative to the likely extent of the species-level diversity in the *L. polytropa* clade.

We provide the most comprehensive sampling to date for members within the *L. polytropa* clade, with over 300 sampled specimens collected from populations across the globe (Figure 2). However, despite the more than 70-fold increase in putative species in the nominal taxon *L. polytropa*, the present sampling effort is likely insufficient to fully capture species-level diversity in this group. For this study, our densest sampling effort targeted a limited number of mountain ranges in western China and the southwestern USA, complemented by opportunistic sampling in other regions. At local scales, for example, individual mountain ranges, high levels of candidate species were recovered (Appendix A). For example, in the La Sal Mountains, in southeastern Utah, USA, up to 14 candidate species were observed. However, candidate species observed on other nearby mountains, e.g., Uinta Mountains (UT, USA), Beartooth Plateau (MT/WY, USA), Saguache Range (CO, USA), had only partly overlapping suites of candidate species (Figure 3). Hence, we speculate that a substantial portion of species diversity in the *L. polytropa* clade remains undiscovered even in the regions most densely sampled for this study. The speculated underestimate of species diversity in the group is further confounded by the fact that vast regions remain poorly sampled, for example, Europe, Northern Asia, and isolated populations such as those in Australasia (Figure 2). Predictive models for estimating species richness, such as that implemented by Lucking et al. [16], can help in providing quantitative estimates of diversity and directing future sampling.

Given the apparent species richness in the group and the unsettled species boundaries, effective strategies must be developed to robustly delimit species boundaries [26,46]. Based on our results, it appears that the ITS coupled with sequence-based species delimitation approaches, such as ASAP [60], is appropriate as a first pass investigation into species boundaries in the *L. polytropa* clade. Following an initial screening of ITS diversity, targeted genome-scale sequencing will be essential to robustly test evolutionary independence among candidate species-level lineages [46,73]. While our multi-locus phylogenetic reconstruction generally resulted in distinct, well-supported clades coinciding with candidate species inferred from ITS sequence data (Figure 3), the multi-locus dataset was insufficient to resolve many backbone-level relationships nor fully corroborate the independence of candidate species. The genome skimming method implemented here provided a wide range of phylogenomic data that also supported rampant unrecognized diversity in the *L. polytropa* clade. By investigating concordance among different DNA datasets [45], e.g., the ITS (standard fungal barcode) and inferences from the mitochondrial and nuclear genomes, we observed a general pattern that suggests long-term, evolutionary independence among the candidate species within the *L. polytropa* clade. Specifically, data from whole-genome sequencing of a subset of the *L. polytropa* group samples support the inference of rampant, unrecognized species-level diversity (Figure 3 and Figure 4), largely congruent with ASAP delimitations. These results suggest that the candidate species in the *L. polytropa* group are not an artifact of limitations due to single-locus species delimitation methods or intragenomic variation within the multi-copy ribosomal tandem repeat [74]. Rather, there are most likely legitimate reproductive barriers resulting in the observed phylogenetic structure and inferred species partitions.

Although genome skimming approaches, such as those implemented here, can provide crucial genome-scale data [48,75,76], more cost- and time-efficient methods, such as target enrichment sequencing [77,78] or restriction-site associated sequencing (RADseq) [26,47,79] might be more appropriate for this clade given the high number of samples that will likely need to be included. Ultimately, genome-scale data can also be used to reconstruct a robust phylogeny for this group to explore additional evolutionary, phylogeographic, and taxonomic questions. Arguably, genetic data alone is not sufficient for robust delimitations of species boundaries and is best when evaluated in conjunction with other information [80]. While these candidate species provide a powerful framework for taxonomic hypothesis testing, we ultimately support a hypothesis-based, integrative approach to species delimitation [26]. In this study, biased specimen sampling, small sample sizes for most candidate species, and pending detailed anatomical assessments all limit the robustness of the species boundaries inferred here. We hope that these species hypotheses serve as a starting point for integrated approaches required for robust species boundaries and good taxonomy [80,81].

The spectacular species-level diversity and unexpected biogeographic patterns highlight the complex speciation and phylogeographic history of the *L. polytropa* clade (Figure 3). Based on current, limited sampling, we found a mix of a few cosmopolitan lineages and many local/regional endemics. Other lichen-forming fungi have shown similar patterns of complex intercontinental species distributions [82,83,84,85]. Presently, little is known of evolutionary processes that may give rise to reproductive isolation and ultimately speciation among populations in the *L. polytropa* clade. Our results indicate frequent, long-distance dispersal throughout the diversification history of this clade (Figure 3). While most candidate species inferred here were not found across broad, intercontinental distributions, ca. 10% comprised specimens from multiple continents. Future work should attempt to elucidate species distributions and different ecological niches among candidate species and factors influencing ecological specialization, dispersal limitations, etc. For example, some morphologically similar species, such as *L. microloba*, appear to have specialized ecological niches and distributions [36], while our data suggest that others have wide ecological amplitude, resulting in more broadly distributed species. Our hope is that this study provides the impetus for using members of the *L. polytropa* clade as a model to explore phylogeography and speciation in symbiotic fungi.

The results of this study have complex taxonomic implications for *L. polytropa* and closely related species. Given the long, unsettled taxonomic history of *L. polytropa* and associated forms and varieties, a careful revision of the complex taxonomy and available type material will be required to identify which candidate species coincide with *L. polytropa* and other formally described species before naming new species can proceed [26]. This effort may be confounded by the lack of consistent diagnostic features separating evolutionarily distinct lineages, potentially due, in part, to symbiotic interactions [23,86]. Initial phenotypic investigations here failed to reveal consistent taxonomically diagnostic traits, e.g., spore size (Appendix A), corroborating a subset of the distinct candidate species. In some cases, intraspecific phenotype variation/plasticity relating to the thallus morphology, secondary metabolite variation, and spore size was observed. *Lecanora fuscobrunnea*, an Antarctic endemic with distinct apothecia morphology, was found to be closely related to several *L. polytropa* specimens from western North America (Figure 3) and were inferred to be conspecific in several ASAP species partition models (Appendix A). However, given the sparse sample size for most candidate species circumscribed in this study and the limited number of specimens examined, this conclusion should be held only tentatively. In contrast, our results show/corroborate that several morphologically distinct taxa also belong to the *L. polytropa* clade, including *L. chlorophaeodes*, *L. dispersoareolata*, *L. fuscobrunnea*, *L. intricata*, *L. solaris*, *L.* cf. *subcinctula*, and *L. subintricata*. While multiple candidate species also occur in a number of these other nominal taxa, the morphologically distinct groups are generally separated from specimens representing *L. polytropa* (Figure 3). Importantly, we propose that future species descriptions include genetic information in the formal description, as the highly variable morphology is often non-diagnostic, making classification based simply on morphology largely unreliable, as evidenced by two centuries of unresolved species boundaries in the *L. polytropa* clade. Despite the indisputable merits of molecular sequence data in taxonomy, these data are only scarcely used for the formal description of taxa. Novel approaches for DNA-based diagnoses, including diagnostic nucleotide combinations in DNA sequence alignments [87], can be used to provide formal diagnoses for these challenging groups. 

Our study lays the groundwork for elucidating important evolutionary insight and formally recognizing undescribed species diversity and taxonomic diversity for this important cosmopolitan clade of symbiotic fungi.

## Figures and Tables

**Figure 1 jof-08-00490-f001:**
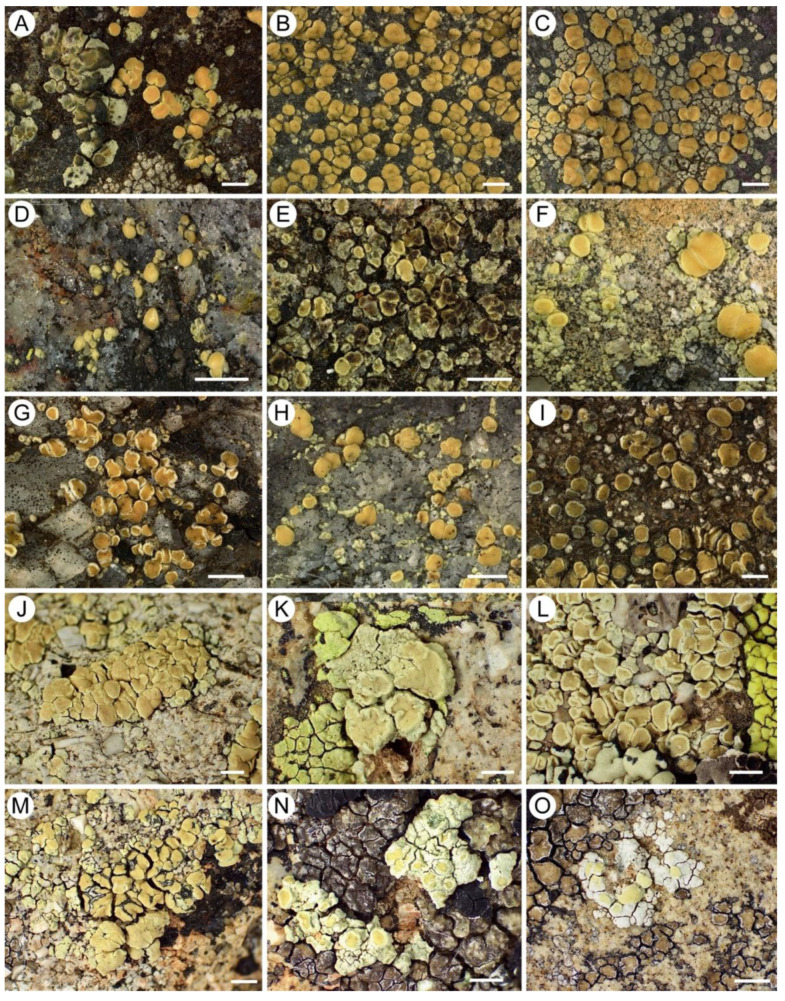
Morphological diversity within the *Lecanora polytropa* group. (**A**) *L. intricata* and *L. polytropa* (McMullin 13834 CANL); (**B**) *L. polytropa* (McMullin 17629 CANL); (**C**) *L. polytropa* (McMullin 17695 CANL); (**D**) *L. polytropa* (McMullin 17811 CANL); (**E**) *L. polytropa* (McMullin 13266 CANL); (**F**) *L. polytropa* (McMullin 22536 CANL); (**G**) *L. polytropa* (McMullin 8820CANL); (**H**) *L. polytropa* (McMullin 22539 CANL); (**I**) *L. polytropa* (McMullin 21121 CANL); (**J**) *L. polytropa* (Leavitt SL18256 BRY-C); (**K**) *L. polytropa* (Leavitt SL18280 BRY-C); (**L**) *L. polytropa* (Leavitt SL18356 BRY-C); (**M**) *L. polytropa* (Leavitt SL18454 BRY-C); (**N**) *L. polytropa* (Leavitt SL18455 BRY-C); and (**O**) *L. polytropa* (Leavitt SL18653 BRY-C). Scale bar = 1 mm.

**Figure 2 jof-08-00490-f002:**
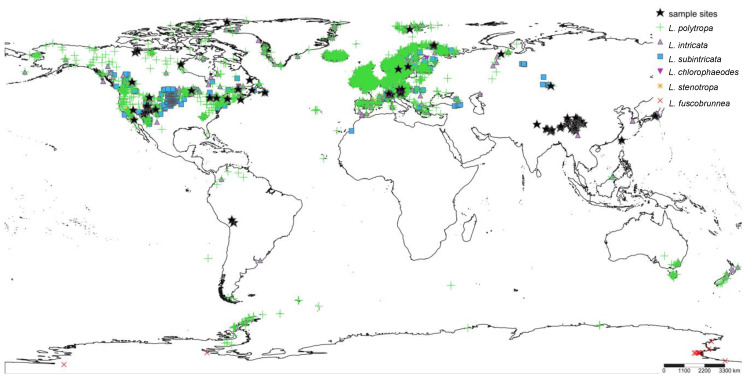
Geographic distribution of species within the *Lecanora polytropa* group based on records available from GBIF (https://www.gbif.org; accessed on 3 February 2022). Sampling sites are indicated with a ‘star’. The locations of specimens representing *L. fuscobrunnea* (Antarctic endemic, downloaded from GenBank) are not shown. The map was procuded using SimpleMappr (https://www.simplemappr.net; accessed on 3 February 2022).

**Figure 3 jof-08-00490-f003:**
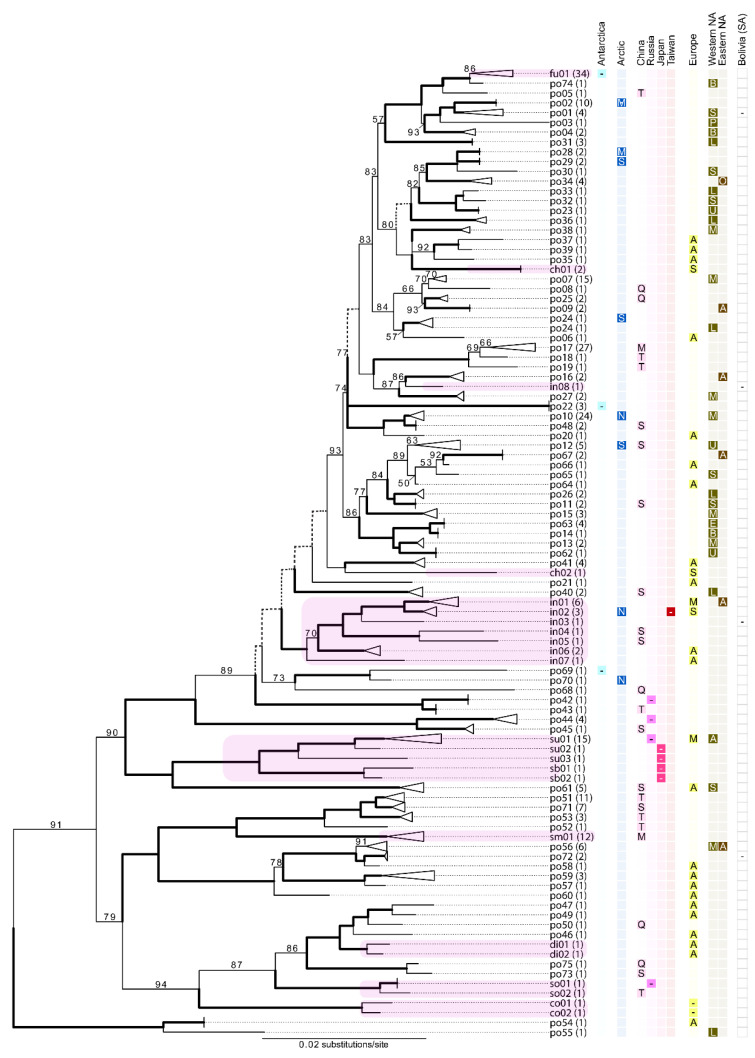
(previous page). Midpoint-rooted, five-marker (ITS, nuLSU, *RPB*1, *RPB*2, mtSSU) maximum likelihood topology of the *L. polytropa* group. Clades are labeled as the candidate species inferred from ITS data using ASAP, with the number of specimens representing each candidate species in parentheses. Thickened black branches indicate bootstrap (BS) values between 95–100%, dashed lines indicate BS values below 50%, and otherwise, bootstrap values are indicated at nodes. Clades highlighted in pink represent other taxa recovered in the *L. polytropa* clade—‘ch’, *L. chlorophaeodes*; ‘co’, *L. concolor*; ‘di’, *L. dispersoareolata*; ‘fu’, *L. fuscobrunnea*; ‘in’, *L. intricata*; ‘sb’, *L.* cf. *subcintula*’; ‘sm’, *L. somervelii*; ‘so’, *L. solaris*; ‘su’, *L. subintricata—*tips labels coincide with names in Appendix A in the ‘Candidate ASAP species partition code’ column. The geographic distribution of each candidate species is shown following the tip label: Antarctica (no distinction for different regions); Arctic (‘N’ = Nunavik; ‘S’ = Svalbard; ‘M’ = multiple locations); China (‘Q’= Qinghai; ‘S’= Sichuan; ‘T’ = Tibet; ‘M’ = multiple locations); Russia (no distinction for different regions); Japan (no distinction for different regions); Taiwan (no distinction for different regions); ‘Western NA’ is western North America (‘B’ = Beartooth Plateau, MT, USA; ‘E’ = Escalante Region, UT, USA; ‘L’ = La Sal Mountains, UT, USA; ‘P’ = Transverse Range, CA, USA; ‘S’= Saguache Range, CO, USA; ‘U’ = Uinta Mountains, UT, USA; ‘M’ = multiple locations); ‘Eastern NA’ is eastern North America (‘M’ = multiple locations); Bolivia, South America (no distinction for different regions). *Carbonea* specimens were recovered as sisters to the remaining samples and not shown—the complete tree is available as Appendix A.

**Figure 4 jof-08-00490-f004:**
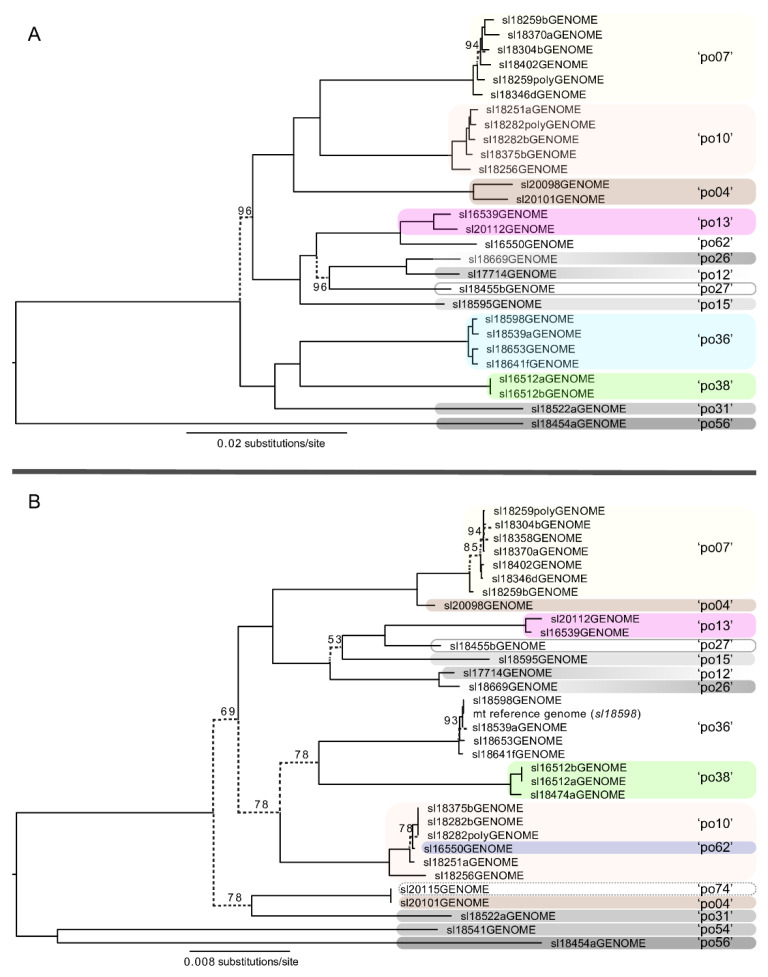
Maximum likelihood (ML) topologies inferred from nuclear (**A**) and mitochondrial (**B**) phylogenomic datasets. (**A**). A ML topology from the concatenated alignments of 1209 BUSCO single copy nuclear markers spanning 2.28 MBps (Appendix A). Colored clades and clade names are linked to the candidate species inferred from the standard DNA barcoding marker for fungi (ITS), and candidate species inferred from the 1209 BUSCO gene trees using SODA are shown with dashed boxes. Only bootstrap values below 100% are shown. (**B**). A ML topology from the 66.5 KBp mitochondrial alignment generated using RealPhy (Appendix A). Only bootstrap values below 100% are shown.

## Data Availability

Newly generated ITS, nuLSU, RPB1, RPB2, and mtSSU sequences are deposited in GenBank under accession numbers ON179980–ON180462 and ON217582–ON217793. Illumina short reads from the 32 ‘*L. polytropa* group’ specimens are available in the NCBI Short Read Archive (PRJNA823672). All alignments are available provided as Appendix A.

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
