# Peer review of "Providing Scale to a Known Taxonomic Unknown—At Least a 70-Fold Increase in Species Diversity in a Cosmopolitan Nominal Taxon of Lichen-Forming Fungi"

_jof, 2022, doi:10.3390/jof8050490_

Round 1
Reviewer 1 Report
Dear Editor:
I reviewed the manuscript entitled ‘Providing scale to a known taxonomic unknown – at least a 70- fold increase in species diversity in a cosmopolitan nominal 3 taxon of lichen-forming fungi’ by Zhang and coauthors.
This study reports an approach to the taxonomy of the Lecanora polytropa group, a problematic group of species occurring in arctic-alpine environments all around the world. The group is known by its wide morphological plasticity what usually makes difficult to assign a reliable name to many specimens collected in those areas.
The approach carried out by the authors include barcoding of the ITS region, sequencing of multiple loci and a metagenomic approach with a subset of specimens. All results point to a previously unknown diversity in the group, which could increase the number of species considerably. The manuscript is well written and use up-to-date literature in the introduction and discussion sections, so no critique can be done in that aspects.
My main concern about the manuscript is the use of ASAP as a single method to provide a species hypothesis and that the hypothesis is not formally tested. I do not doubt about the results obtained, which are very plausible, but I think methodology should be improved to avoid the possibility of spurious results. ASAP tends to largely oversplit taxa, especially when used in a group of closely related species and not for a large inventory of non-related taxa. I would suggest the use of other algorithms for single-locus species delimitations to test ASAP results. Further, considering the spirit of the integrative taxonomy appealed in the introduction section, the hypothesis obtained from the single-locus delimitation should be tested against other hypotheses, for instance one derived from morphological observations (general growth form), or classes derived from ascospores sizes, or previous specimen classification based on traditional keys and taxonomic information. Concordance among molecular regions is not a test per se, and concordance is not a proof of the presence of true species.
Finally, I would like to see in the discussion section some considerations about the reliability of the results, if there may exist evolutionary scenarios that would confound the results.
Author Response
see- attached

Reviewer 2 Report
Overall
The manuscript reports the phylogenetic and phylogenetic investigation into the species circumscription of the Lecanora polytropa complex, one of the most difficult species complex in a complicated genus like Lecanora. The authors have recovered a staggering degree of species diversity within this complex. The manuscript is generally well written, and the authors used appropriate tools for the questions at hands. Multiple types of datasets have been used here, but they were always clearly explained why they were being used in slightly different ways. I am sure that many of the discrepancies were due to technical difficulties and limitations on access to specimens, but it would be nice to elaborate more on the use of these different datasets to achieve different objectives of this study. While no immediate phenotypes could be detected among candidate species, the study will certainly pave the way for more detailed taxonomic revisions in the future.
minor suggestion
- L.23-27 "For a subset of the ..., four-marker data set." It's bit hard to understand what each of the three (1209 single-copy genes, ASAP, four marker dataset). analyses did in the overall framework. Maybe rephrase this a bit for clarity.
- L. 136-137. It is not very clear what it means by "local patterns of diversity." Please elaborate more on this point.
- L. 145. "-see Results" here seems out of place.
- L. 146. Why were only the specimens from western USA observed for the diagnostic phenotypes? At least general growth forms can be observed from herbarium specimens.
- L. 165: "For selected specimens representing the genetic diversity" > how were these specimens selected? Did they represent each clade in the larger Lecanora phylogeny?
- L. 182: How were these 32 specimens selected? What did we hope to gain from doing the Illumina's metagenomic reads from the specimens that we could not tell from the Sanger's dataset?
- L. 249: Why did we use the 2-locus dataset to assess the monophyly of "L. polytropa group, when we already have the other 5-locus dataset already?
- L. 254: "see Results" seem out of place again. Maybe better in parentheses?
- L. 346-347: "Based on ..." > not clear what this sentence is trying to say.
- L. 368: It would be nice to see the comparison with the other large species complexes for lichens for the scale.
Figures and others
- Figure 2: The species names in the legend should be italicized.
- Figure 3: Is it the figure on page 9 or the figure on the following page? The Latin names should be italicized in the caption.
- Figure 4: 4A and 4B are not clearly labeled on the figure. Also one bootstrap value at 100 is also shown in 4B?
- The "candidate species names" are different between Figure 3 and Figure 4. It's understandable that the first two letters at the tips of Figure 3 are short-hand for the species group. So maybe use the same abbreviation for the Figure 4?
- How is Appendix different from Supplementary Materials ?
